# Metabolomics-Based Analysis of the Major Taste Contributors of Meat by Comparing Differences in Muscle Tissue between Chickens and Common Livestock Species

**DOI:** 10.3390/foods11223586

**Published:** 2022-11-11

**Authors:** Yanke Wang, Xiaojing Liu, Yongli Wang, Guiping Zhao, Jie Wen, Huanxian Cui

**Affiliations:** State Key Laboratory of Animal Nutrition, Key Laboratory of Animal (Poultry) Genetics Breeding and Reproduction, Ministry of Agriculture, Institute of Animal Science, Chinese Academy of Agricultural Sciences, Beijing 100193, China

**Keywords:** metabolomics, taste contributors, muscle tissues, e-tongue

## Abstract

The taste of meat is the result of complex chemical reactions. In this study, non-target metabolomics was used to resolve the taste differences in muscle tissue of four major livestock species (chicken, duck, pork, and beef). The electronic tongue was then combined to identify the major taste contributors to meat. The results showed that the metabolism of chicken meat differed from that of duck, pork, and beef. The multivariate statistical analysis showed that the five important metabolites responsible for the differences were all related to taste, including creatinine, hypoxanthine, gamma-aminobutyric acid, L-glutamic acid, and L-aspartic acid. These five key taste contributors acted mainly through the amino acid metabolic pathways. In combination with electronic tongue (e-tongue) analysis, inosine monophosphate was the main contributor of umami. L-Glutamic acid and L-aspartic acid might be important contributors to the umami richness. Creatinine and hypoxanthine contributed more to the bitter aftertaste of meat.

## 1. Introduction

Meat has a distinct position in the food basket and has been widely consumed by humans since the pre-historic era because it is a ready energy source with high-quality proteins, palatability, strength, and power [1]. The production and consumption of meat in China have both increased rapidly over the years, and pork, chicken, duck, and beef consumption have been at the top of the list [2]. However, the long-term growth selection and the emergence of high-density intensive farming patterns have led to the decline in the meat quality of livestock and poultry to meet the demand for meat quantity [3]. However, high-quality meat is increasingly in demand by consumers, especially meat with rich taste [4].

The flavor of meat is the aroma and taste of meat produced by a series of chemical changes in the flavor precursors in raw meat during processing [5]. The aroma is mainly produced by volatile organic compounds [6]. The taste mainly includes five major tastes, umami, salty, sour, bitter, and astringent, which is mainly caused by water-soluble taste substances, such as sugars, inorganic salts, amino acids, nucleotides, lactic acid, and peptides [7]. A lot of research has been conducted on the flavor of major livestock and poultry meat to improve the quality of livestock products. Studies have shown that free amino acids are highly correlated with the taste of chicken [8]. Inosine monophosphate (IMP), a major umami nucleotide substance, has become an important indicator for evaluating meat umami [9]. Additionally, IMP has received increasing attention due to its unique taste and important application as a taste enhancer in various foods [10]. Acetic acid and butyric acid were associated with the sour taste of duck meat [11]. The special taste of pork was mainly due to the important components in meat, such as sodium glutamate and IMP [12]. Glutamate and IMP were also taste-active components of chicken, and a synergistic effect is observed between glutamate and IMP [13]. Studies comparing the biochemical indicators of chicken, duck, pork, and beef are conducted to determine whether beef is adulterated [14]. However, studies have focused on the taste substances of individual species, and a few comparative studies have been performed on the taste differences among the major livestock species.

Modern and high-efficiency analytical instruments and separation techniques are required for their determination due to the complexity of the taste components and the minute content [15]. Metabolomics is a high-resolution, high-sensitivity, and high-throughput technique based on methods, such as mass spectrometry or nuclear magnetic resonance. It has been widely used in meat quality evaluation to screen potential biomarkers associated with meat quality traits [16]. Recently, a study based on metabolomics to monitor the changes in beef quality after 0, 1, 10, 17, and 44 days of storage at 1 °C found that glutamic acid, serine, and arginine could be used as parameters or indicators of meat taste [17]. In addition, the metabolomics data suggested that IMP was a metabolic signature and biomarker among frozen meats during refrigeration [18]. In addition, the electronic tongue (e-tongue) can evaluate the taste of food objectively, quickly, and simply, and it has been widely used in food, medicine, tea, and wine industries [19]. However, the combined use of metabolomics and e-tongue to identify the major taste contributors has not been reported.

This study was conducted with chicken, duck, pork, and beef at market age. The untargeted metabolomics of meat samples was studied by liquid chromatography–tandem mass spectrometry (LC-MS/MS). The multivariate statistical analysis was used to screen and identify the key metabolites that caused the difference in meat taste among different livestock species. Then, a Kyoto Encyclopedia of Genes and Genomes (KEGG) analysis of the differential metabolites was performed to screen out common metabolic pathways. Finally, the major taste contributors were identified in combination with the e-tongue. This study could provide a theoretical reference for the in-depth study of meat taste.

## 2. Materials and Methods

### 2.1. Animals and Sample Preparation

This study was conducted following the Guidelines for using Experimental Animals, established by the Ministry of Science and Technology (Beijing, China). All experimental protocols were approved by the Science Research Department of the Institute of Animal Sciences, Chinese Academy of Agricultural Sciences (Beijing, China) (No. IAS2019-21).

The chicken (Wenchang, 98-day-old, female), cattle (Huaxi, 1-year-old, female), swine (large white pork, 6-month-old, female), and duck (Peking duck, 8-week-old, female) muscle meats were provided and appraised by the Beijing Institute of Animal Science, Chinese Academy of Agricultural Sciences. Five muscle tissue samples of female chicken (breast muscle tissue), duck (breast muscle tissue), pork (longissimus dorsi), and beef (longissimus dorsi) were selected. All muscle samples were trimmed with sterile scissors to remove visible fascia, fat, and connective tissue. The 20 muscle tissue samples for e-tongue and metabolomic experiments were stored at −80 °C.

### 2.2. E-Tongue Analysis

The e-tongue analysis was performed using the e-tongue device SA402B, produced by INSENT Corporation of Japan. For this, 5 g meat samples were weighed into polyethylene plastic bags, sealed, and placed in a water bath at 100 °C for heating for 30 min. Then, the boiled meat samples were put in a 50 mL centrifuge tube, mixed with 20 mL of ultra-pure water preheated in a 37 °C water bath, vortexed for 30 min, mixed with 20 mL of ultra-pure water (37 °C) again, and sonicated for 30 min to promote the full dissolution of water-soluble taste substances in water. The samples were centrifuged at 10,000 rpm for 10 min, and the supernatant was filtered with a sand core funnel to obtain a clear filtrate without precipitation. The filtrate was diluted two times to obtain a sample extract, which was evenly transferred to two specific sample cups to determine the e-tongue. The e-tongue assay was started after calibration of the sensor and activation of the electrode with a reference solution (30 mM KCl + 0.3 mM tartaric acid) for 24 h. During the test, the sensor was washed with a cleaning solution (30 mM KCl + 0.3 mM tartaric acid, 90 s) and a reference solution (30 mM KCl + 0.3 mM tartaric acid, 120 s + 120 s), and the sample solution was immersed for 30 s for detection. Three individual samples from each species of chicken, duck, pig, and cow were selected for e-tongue measurement, and each sample was measured four times. The data of the last three cycles were selected for statistics and analysis to ensure the stability of the data.

### 2.3. LC-MS/MS Analysis

The samples were transferred to an Eppendorf tube. After adding extract solution (acetonitrile: methanol = 1:1, containing isotopically labeled internal standard mixture), the samples were vortexed for 30 s, sonicated for 10 min in an ice-water bath, and incubated for 1 h at −40 °C to precipitate proteins. Then, the samples were centrifuged at 12,000 rpm for 15 min at 4 °C. The resulting supernatant was transferred to a 2 mL LC/MS glass vial for UHPLC-QE-MS analysis. The quality control sample was prepared by mixing an equal aliquot of the supernatants from all the samples. LC-MS/MS analyses were performed using a UHPLC system (Vanquish, Thermo Fisher Scientific, Waltham, MA, USA) with a UPLC BEH amide column coupled to a Q Exactive HF-X mass spectrometer (Orbitrap MS, Thermo, Waltham, MA, USA). The mass spectrometer was used for its ability to acquire MS/MS spectra in information-dependent acquisition modes in the control of the acquisition software (Xcalibur, Thermo). The acquisition software continuously evaluated the full-scan MS spectrum in this mode.

### 2.4. Data Treatment and Pre-Processing

The raw data were converted into mzXML format using ProteoWizard and developed in R language, based on XCMS for peak detection, extraction, alignment, and integration. Then, an in-house MS2 database (BiotreeDB) was applied in metabolite annotation [20].

Positive and negative ion modes were used to detect metabolites, increasing metabolite coverage and improving detection.

### 2.5. Statistical Analysis

The multivariate analyses, including principal component analysis and orthogonal partial least squares discriminant analysis (OPLS-DA), were performed using GENE DENOVO Company Metabolic Website (https://www.omicshare.com/, accessed on 3 February 2022). Variable importance in projection (VIP) was performed, and the metabolites with a VIP value ≥ 1 (*p* < 0.05) were regarded as the most influential differential metabolites in the extracted OPLS model. The metabolite pathway involved in differential metabolites was constructed according to an analysis of the KEGG database. The significance of important metabolites was analyzed with SPSS 26.0 statistical software (IBM Inc., Chicago, IL, USA) using analysis of variance at the *p* < 0.05 level. GraphPad Prism 8.0 software (GraphPad Software Inc., San Diego, CA, USA) was used to generate graphs.

## 3. Results

### 3.1. Analysis of the Differences in Individual Tastes of Muscle Tissue of Four Different Livestock Species

We evaluated the taste of muscle tissue of different livestock species using e-tongue. We analyzed umami, umami richness, bitterness, bitterness aftertaste, sourness, saltiness, astringency, and astringency aftertaste to understand more precisely which individual taste was responsible for the taste differences in the muscle tissue of different livestock species. As shown in Figure 1, the umami, bitterness, sourness, saltiness, astringency, and astringency aftertaste of chicken, duck, pork, and beef were not different, indicating that the e-tongue could not distinguish these individual tastes of the four animal species. However, the umami richness and bitter aftertaste of chicken were independent of the other three species, while the umami richness and bitter aftertaste of duck, pork, and beef were intertwined, indicating that the e-tongue could distinguish the umami richness and bitter aftertaste of chicken from the other three animal species and could not distinguish these two individual tastes of duck, pork, and beef.

### 3.2. Clustering Analysis of Metabolites in Muscle Tissues of Four Different Livestock Species

We performed metabolome sequencing on five individuals of each species to search for relevant taste metabolites that contribute to the umami richness and bitter aftertaste that distinguishes chicken from the other three livestock species. A total of 6272 molecular signature peaks were detected by LC-MS/MS in the positive ion mode from chicken, duck, pork, and beef. Of these, 1655 metabolites were annotated. Additionally, a total of 5326 molecular signature peaks were detected in the negative ion mode, of which 970 metabolites were annotated.

The samples of the four livestock species were analyzed by hierarchical clustering to form a clustering tree, showing the similarity between samples, and samples within a cluster had a higher inter-sample similarity. As shown in Figure 2A, the hierarchical clustering analysis found that, in the positive ion mode, chicken and duck meat were clustered, and pork and beef were clustered, indicating that the poultry samples were similar, the livestock samples were similar, but the metabolism between poultry and livestock was different. However, the three livestock species of duck, pork, and beef clustered in the negative ion mode (Figure 2B), indicating that duck, pork, and beef were similar between samples in the negative ion mode and differed from chicken samples. The hierarchical clustering results in the negative ion mode were consistent with the results of the umami richness and bitterness aftertaste of the electronic tongue, which both indicated that chicken was different from duck, pork, and beef. Therefore, the clustering results in the negative ion mode were more closely related to the taste differences of the four livestock meats.

### 3.3. Analysis of Metabolic Differences between Chicken Versus Duck, Pork, and Beef

The results of the electronic tongue showed that the umami richness and bitter aftertaste of chicken were different from those of duck, pork, and beef. We first performed OPLS-DA analysis on the metabolites of the three groups of chicken versus duck, chicken versus pork, and chicken versus beef to find the metabolites responsible for the taste differences. OPLS-DA combined the two methods of orthogonal signal correction and partial least squares discriminant analysis (PLS-DA), which could decompose the X matrix information into two types of information related to Y and irrelevant information, before filtering the difference variables by removing the irrelevant differences. Subsequent model testing and differential metabolite screening were analyzed using OPLS-DA results. As shown in Figure 3, OPLS-DA could distinguish chicken from duck, chicken from pork, and chicken from beef in positive and negative ion modes, indicating differences in metabolites between chicken and duck, pork, and beef.

### 3.4. Screening for Important Taste Substances Contributing to the OPLS Model

The loading plot identified the variables that contributed the most to the principal components of the OPLS model. The variables that were far from the origin in the horizontal coordinate direction contributed more to the differentiation of the two groups of samples. We listed the 10 variables separately that contributed more to distinguish chicken from duck, pork, and beef (Figure 4). The 10 variables in each of the three groups had both similarities and differences. In addition, nine common variables (POS05070, NEG02984, POS00024, NEG00029, NEG01334, NEG04825, NEG04290, POS02452, and POS05044) contributed significantly to the differentiation of all three groups, chicken versus duck, chicken versus pork, and chicken versus beef, of which two substances were annotated: creatinine (POS00024) and hypoxanthine (NEG00029).

### 3.5. KEGG Analysis of Differential Metabolites and Comparison of Important Taste Substances

VIP values were used to illustrate the importance of variables (eigen peaks) that explained the X dataset and the associated Y dataset. When the VIP of a variable was greater than 1, it indicated that the variable was important. Therefore, it was used as one of the screening conditions for potential biomarkers. Therefore, a combination of multivariate statistical analysis of VIP values of OPLS-DA (VIP ≥ 1) and univariate statistical analysis of t test *p* values (*t*-test *p* < 0.05) was used to screen for differential metabolites between the different comparison groups. KEGG analysis was performed on each group of differential metabolites to explore through which pathways the differential metabolites acted. The results showed that the differential metabolites of the three groups involved eight common pathways, and four of the eight pathways were related to amino acids, including the histidine metabolism pathway, GABAergic synapse pathway, alanine, aspartate, and glutamate metabolism pathway, and protein digestion and absorption pathway (see Table 1).

In addition, three major metabolites were involved in the eight pathways. Among the three major metabolites, gamma-aminobutyric acid was the major metabolite through neuroactive ligand–receptor interaction, GABAergic synapse, nicotine addiction, and alanine, aspartate, and glutamate metabolism. L-glutamic acid was the major metabolite through the remaining seven pathways, except cyanoamino acid metabolism, and L-aspartic acid was the major metabolite through the remaining six pathways, except GABAergic synapse and nicotine addiction. A total of five important taste substances were identified by load map and KEGG analysis, and the level of these substances differed significantly among the four livestock species. As shown in Figure 5, the contents of hypoxanthine, gamma-aminobutyric acid, L-glutamic acid, and L-aspartic acid in chicken were significantly higher than those of the other three species. However, the creatinine content of pork was significantly higher than that of chicken, duck, and beef.

## 4. Discussion

The quality and taste of meat are important factors for people when choosing meat products [21]. To date, most studies have identified taste compounds in chicken, duck, pork, and beef by high-throughput assay techniques, such as GC-MS [22]. However, fewer studies have been conducted to analyze the taste differences in meat from different livestock species from a metabolomic perspective. IMP, aspartic acid, glutamic acid, and many other substances are related to meat taste, but which taste these substances contribute to is not clear [23]. Compared with the previous use of a single species, this study performed a multivariate statistical analysis based on the metabolomics of three groups, chicken versus duck, chicken versus pork, and chicken versus beef, to identify important taste metabolites, followed by a combination of e-tongue analysis to identify the important contributors to umami and bitterness.

E-tongue showed that the umami richness and bitter aftertaste of chicken were different from those of duck, pork, and beef. The loadings plot helped identify the metabolites that contributed the most to the variation in metabolite patterns between the comparison groups [24]. In this experiment, we used the loading plot to find the taste metabolites that differentiated chicken from the other three species. Two common metabolites, creatinine and hypoxanthine, were identified by load diagrams as important contributors to distinguish chicken from duck, pork, and beef. Both substances were associated with taste presentation; creatinine was related to bitterness [25]. As a metabolite of IMP, excess hypoxanthine led to the production of the bitter taste of meat [26]. The feasibility of such a grouping to find the taste metabolites responsible for the differences was confirmed by comparing chicken with the remaining three species screened for common important contributors associated with taste.

We also identified gamma-aminobutyric acid (GABA), L-glutamic acid, and L-aspartate acid as the main metabolites that contributed to the taste difference between chicken and duck, pork, and beef and mainly acted through the amino acid metabolism pathways. Glutamic acid contributed to meat taste, including “umami” and “brothy” descriptors, and was one of the important taste-active components of meat [27]. L-Glutamic acid and L-aspartic acid were substances that made an important contribution to umami [28,29]. In addition, the physiological activity of GABA had the physiological functions of regulating blood pressure, promoting mental stability, and nourishing nerve cells, which had no direct effect on taste [30], but the main pathway of GABA synthesis was the decarboxylation of the umami taste-contributing substance, L-glutamic acid, which also has an indirect relationship with taste [31]. Both the loading plot results and the KEGG results of the differential metabolites indicated that chicken meat was different from duck, pork, and beef, mainly due to umami and bitterness taste substances, which was consistent with the results of the e-tongue.

The e-tongue is an instrument based on high-sensitivity sensors, which can distinguish food through pattern recognition and evaluate various tastes of food, such as common umami, salty, and sour taste [32]. IMP is the main umami substance of meat, which is usually used as an indicator of meat umami [33]. The e-tongue results showed that the umami of chicken meat did not differ from those of the other three livestock species. Additionally, the content of IMP in chicken meat did not differ significantly from that in the other three livestock species (Figure 5A). The differences in the content of IMP among animal species were consistent with the results of the e-tongue to detect the differences in the umami taste of different animal species. Therefore, we speculated that IMP was the main contributing substance to umami, in agreement with previous findings [34]. The e-tongue tests showed that the taste richness of chicken was different from that of the other three species. The contents of L-glutamic acid and L-aspartic acid in chicken were significantly higher than those in duck, pork, and beef. The differences in the contents of L-glutamic acid and L-aspartic acid were consistent with the differences in umami aftertaste detected by the e-tongue among animal species. Studies showed that L-glutamic acid and L-aspartic acid were also substances contributing to umami [35]. Therefore, we speculated that the contribution of L-glutamic acid and L-aspartic acid to umami was mainly to enhance umami richness. Maughan et al. reported that chicken and pork had significantly higher umami compared with beef; our results also showed that chicken umami was stronger than that of the other three species [36]. The bitter aftertaste of chicken was different from that of the other three species (Figure 1D), and the contents of creatinine and hypoxanthine in chicken were also significantly different from those in the other three species. Both creatinine and hypoxanthine were related to the bitter taste of meat [37], and the differences in the contents of creatinine and hypoxanthine were consistent with the differences in the bitterness aftertaste between animal species detected by the e-tongue. Therefore, we speculated that creatinine and hypoxanthine were the main contributors to the bitter taste by enhancing bitter aftertaste.

Based on this finding, our results established that chicken meat differed in umami richness and bitter aftertaste from duck, pork, and beef. In addition, important contributors to the main tastes (umami, umami richness, bitter aftertaste) were identified after a comprehensive analysis. However, the molecular mechanism of important taste-contributing substances needs further exploration. Therefore, the results of this study indicated that we needed to use the combination of metabolomics and transcriptomics in the future to identify the developmental patterns and potential mechanisms of taste substances that caused differences.

## 5. Conclusions

In summary, the taste differences between chicken and common livestock species were systematically compared based on metabolomics, and the main taste contributors were identified in combination with e-tongue. Five key metabolites of the taste were identified by multivariate statistical analysis, including creatinine, hypoxanthine, gamma-aminobutyric acid, L-glutamic acid, and L-aspartic acid. The results of KEGG suggested that these five key taste contributors acted mainly through amino acid metabolic pathways, leading to taste differences between chicken meat and meat from the other three livestock species. Additionally, the contents of four of these five substances in chicken meat were significantly different from that in duck, pork, and beef. Combined with the results of the e-tongue, the study further showed that IMP was the main contributor to umami, while L-glutamic acid and L-aspartic acid might be the main contributors to umami richness. Additionally, creatinine and hypoxanthine contributed to bitter aftertaste. This study provided a scientific basis for better control of meat quality and development of methods to improve meat quality.

## Figures and Tables

**Figure 1 foods-11-03586-f001:**
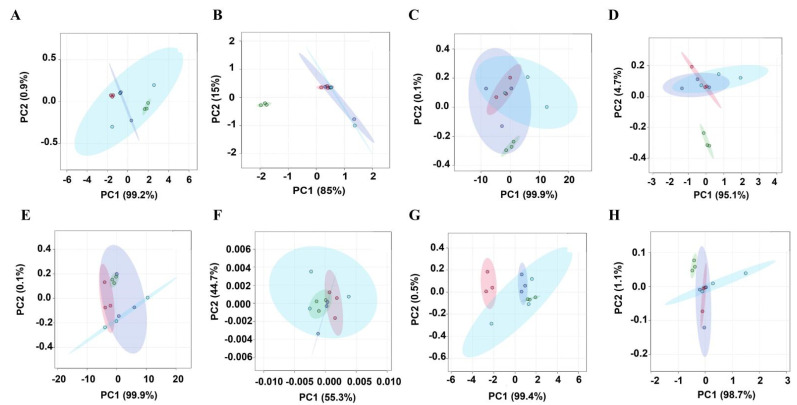
Identification of individual taste differences in meat from four livestock species using an e-tongue. In the picture, green represents chicken, purple represents duck, blue represents pork, and pink represents beef. Differences in umami (**A**), umami richness (**B**), bitterness (**C**), bitterness aftertaste (**D**), sourness (**E**), saltiness (**F**), astringency (**G**), and astringency aftertaste (**H**) were identified for the four animal species, respectively.

**Figure 2 foods-11-03586-f002:**
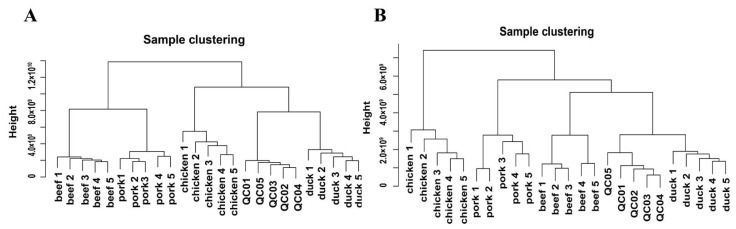
Hierarchical clustering analysis of metabolism of four livestock species. (**A**) Hierarchical clustering analysis of chicken, duck, pork, and beef in the positive ion mode. (**B**) Hierarchical clustering analysis of chicken, duck, pork, and beef in the negative ion mode.

**Figure 3 foods-11-03586-f003:**
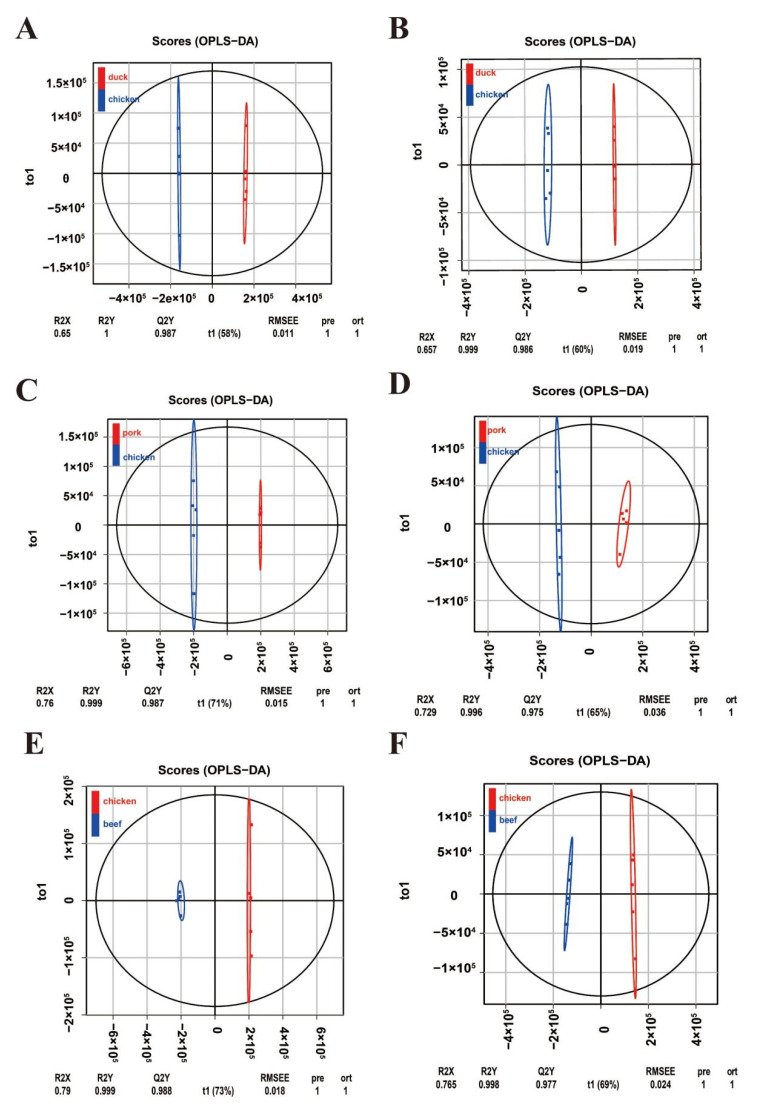
Orthogonal partial least squares discriminant analysis (OPLS-DA) of metabolites between groups. Metabolic differences between chicken and duck meat under positive ion (**A**) and negative ion (**B**) modes. Metabolic differences between chicken and pork under positive ion (**C**) and negative ion (**D**) modes. Metabolic differences between chicken and beef under positive ion (**E**) and negative ion (**F**) modes.

**Figure 4 foods-11-03586-f004:**
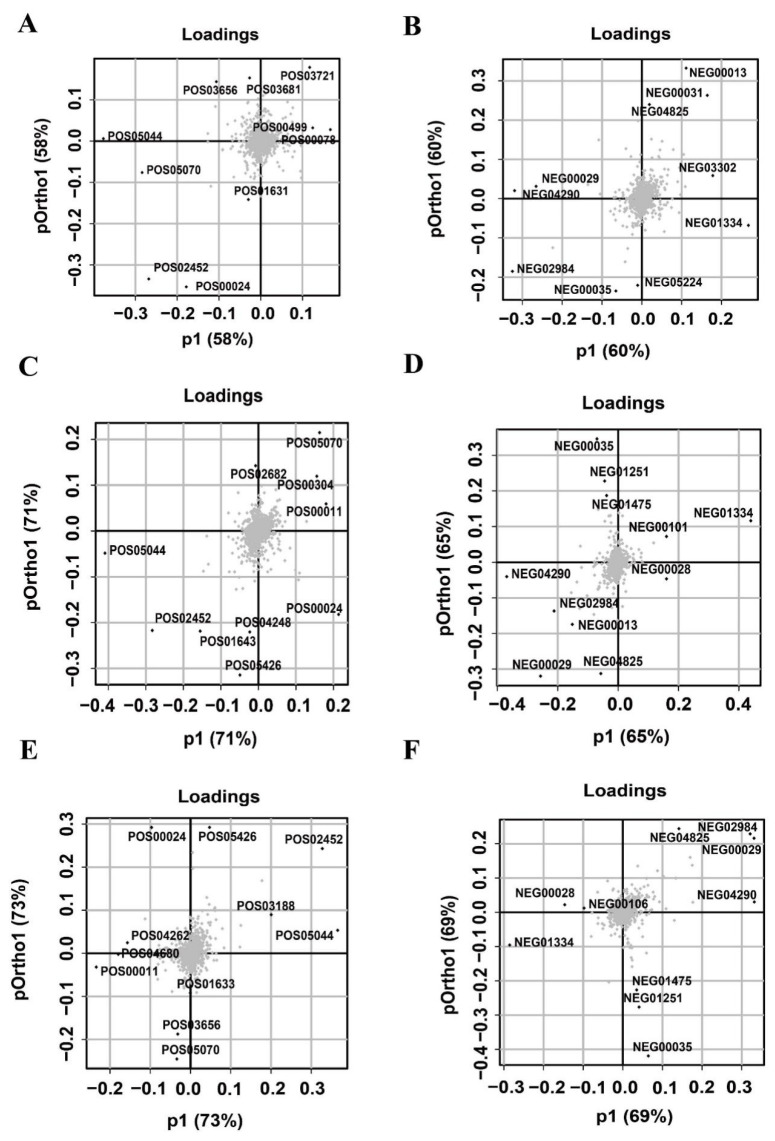
Screening of metabolites with high contribution to the differentiation of groups using the loading plot. Important variables that distinguish chicken and duck meat in positive ion (**A**) and negative ion (**B**) modes. Important variables that distinguish chicken and pork in positive ion (**C**) and negative ion (**D**) modes. Important variables that distinguish chicken and beef in positive ion (**E**) and negative ion (**F**) modes.

**Figure 5 foods-11-03586-f005:**
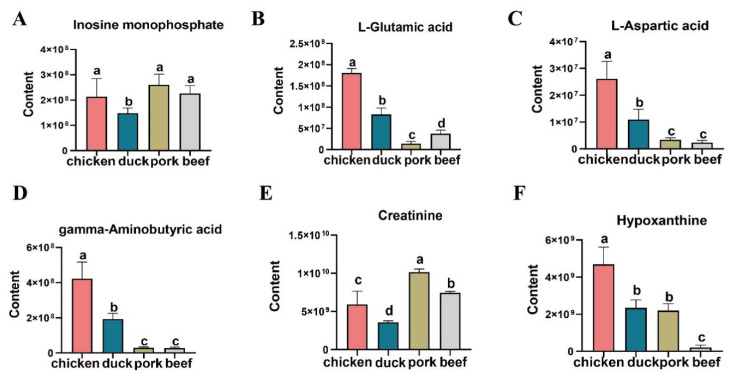
Content of key taste contributors in chicken, duck, pork, and beef. (**A**–**F**) Contents of inosine monophosphate, L-glutamic acid, L-aspartic acid, gamma-aminobutyric acid, creatinine, and hypoxanthine in four livestock species, respectively. The same letter in the figure indicates no significant difference in the content of taste metabolites between species, and different letters indicate a significant difference in the content of taste metabolites between species.

**Table 1 foods-11-03586-t001:** Common pathways of the three groups and the taste metabolites involved in the pathways.

Pathway	ID	Name
Histidine metabolism	POS00077	3-Methylhistidine
	POS00395	Formiminoglutamic acid
	NEG00068	3-Methylhistidine
	NEG00083	L-Glutamic acid
	NEG00088	Anserine
	NEG00241	L-Aspartic acid
	NEG02183	-
Neuroactive ligand–receptor interaction	POS00078	Taurine
	POS00195	Gamma-aminobutyric acid
	POS00505	2-Acetyl-1-alkyl-sn-glycero-3-phosphocholine
	NEG00060	Gamma-aminobutyric acid
	NEG00083	L-Glutamic acid
	NEG00241	L-Aspartic acid
GABAergic synapse	POS00195	Gamma-aminobutyric acid
	NEG00060	Gamma-aminobutyric acid
	NEG00083	L-Glutamic acid
Nicotine addiction	POS00195	Gamma-aminobutyric acid
	NEG00060	Gamma-aminobutyric acid
	NEG00083	L-Glutamic acid
Alanine, aspartate, and glutamate metabolism	POS00195	Gamma-aminobutyric acid
	POS04633	-
	NEG00060	Gamma-aminobutyric acid
	NEG00083	L-Glutamic acid
	NEG00241	L-Aspartic acid
Protein digestion and absorption	NEG00083	L-Glutamic acid
	NEG00241	L-Aspartic acid
ABC transporters	POS00078	Taurine
	NEG00083	L-Glutamic acid
	NEG00241	L-Aspartic acid
Cyanoamino acid metabolism	NEG00241	L-Aspartic acid

## Data Availability

All relevant data supporting the findings of this manuscript are provided in the manuscript; further raw data will be provided from the corresponding author upon reasonable request.

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
