# Peer review of "Metabolomics-Based Analysis of the Major Taste Contributors of Meat by Comparing Differences in Muscle Tissue between Chickens and Common Livestock Species"

_foods, 2022, doi:10.3390/foods11223586_

Round 1

Reviewer 1 Report

The article "Metabolomics-based analysis of taste differences and major 1 taste contributors in muscle tissues of different livestock species" of Wang et al. describe metabolomics methodology for identify differences between the taste of muscle tissues from different animals.

The manuscript was well organized and the use of electronic tongue provided valuable information to the research.

In my opinion the manuscript need minor revision of the figures, because the quality of them are really poor and it is really complicate to read the information in them. The words are too small.

Author Response

Thank you very much for your suggestions on the manuscript. And we have followed your suggestion to enlarge the words on all the pictures in the manuscript. The specific changes can be found in the document "revisions-manuscript".

Reviewer 2 Report

Authors tried to find metabolomics differences and taste relevant factors of four livestocks.

Major comments

1.       Authors compared chicken with three different meats one by one. Why didn’t the authors compare the four different meats in parallel. This analysis made this manuscript finding different metabolites in chicken as compared to three meats. If the authors didn’t intend to compare chicken with the others, re-analyze the four meats in parallel.

2.       What does the figure 4 means? What is the meaning of the distance from center? Please use higher resolution for this figure.

3.       In line 163-166: do authors mean 6272 metabolites? Or number of peaks?

Author Response

Thank you very much for your advice. We have revised the text as you suggested and uploaded the revised version (revisions-manuscript) of the document and the instruction file (cover letter).

  1. Authors compared chicken with three different meats one by one. Why didn’t the authors compare the four different meats in parallel. This analysis made this manuscript finding different metabolites in chicken as compared to three meats. If the authors didn’t intend to compare chicken with the others, re-analyze the four meats in parallel.

Response: Thank you very much for your advice. The reason we chose to compare chicken with three different types of meat (duck, pork, and beef) instead of analyzing all four meats in parallel is: the results of the e-tongue in section 3.1 of the article showed that the umami richness and bitter aftertaste of chicken were different from duck, pork and beef; the results of the hierarchical clustering in the negative ion model in section 3.2 showed that duck, pork and beef clustered together, while chicken differed from these 3 meats. Both the results of the e-tongue and the hierarchical clustering results in the negative ion condition indicated that chicken meat was different from the other three meats, so we used this result as a basis for the subsequent comparative analysis of chicken meat and the three different meats.

  1. What does the figure 4 means? What is the meaning of the distance from center? Please use higher resolution for this figure.

Response: Thank you very much for your suggestions. Figure 4 shows the top 10 variables that contribute most to the principal components of the OPLS model for the 3 groups, i.e. the important variables that cause chicken to be different from duck, different from pork and different from beef.

Distance from center is the distance from the origin of these important variables in the horizontal coordinate direction, with variables away from the origin making a greater contribution to distinguishing the two groups of samples. These are described in detail in lines 210-215 of the text.

Thank you for pointing this out, we have showed higher resolution images in Figure 4.

  1. In line 163-166: do authors mean 6272 metabolites? Or number of peaks?

Response: Thank you very much for your suggestion. We have checked the data again according to your suggestion and found that your statement is correct. LC-MS/MS detected 6272 molecular characteristic peaks, but we incorrectly described it as 6272 metabolites in the article. We have corrected this in lines 164 and 167 of the article.

Reviewer 3 Report

This paper was well written and a great job conforming to the English language usage throughout.  I have made a few written comments throughout the paper and is attached for your review and corrections.  I have noted a few word changes, deletions, etc. and mainly suggest an enlargement of a few of the figures to make for easier examination and interpretation.  Also a couple of places to more clearly identify the species comparisons in the figures would be most helpful.  Minor edits.  Very good job.   Please see the entire document with comments below.   

Author Response

This paper was well written and a great job conforming to the English language usage throughout.  I have made a few written comments throughout the paper and is attached for your review and corrections.  I have noted a few word changes, deletions, etc. and mainly suggest an enlargement of a few of the figures to make for easier examination and interpretation. Also a couple of places to more clearly identify the species comparisons in the figures would be most helpful.  Minor edits.  Very good job.   Please see the entire document with comments below.   

Response: Thank you very sincerely for your suggestions on my manuscript and we have listed the issues mentioned in your review document below. We have revised the manuscript in its entirety in accordance with your suggestions.

  1. Which muscle of the back on beef and pork was sampled? Need to specify above. (longissimus dorsi)?

Response: Thank you very much for your suggestions. We apologise for the lack of clarity. We conducted experiments using the longissimus dorsi of pork and beef. We have made the suggested changes in lines 85-86 of the text.

  1. Were any analysis for fat content of each specie samples conducted? If so, should be stated. If not, a reiteration of trimming to remove all visible fat and connective tissuewas performed. Fat is a major flavor contributor.

Response: Thank you very much for the reminder. We agree with your statement that fat is a major flavour contributor. However, we did not analyse the fat content of each sample. Therefore, we have removed the obvious visible fat and connective tissue from the samples during pre-treatment to prevent it from interfering with the results. We may not have detailed the sample pre-treatment process in the text, so we have made the changes in lines 86-87 of the article.

  1. List company name  

Response: Thank you very much for the heads up. We have added the name of the company on lines 131-132 as you reminded us.

  1. Suggest enlarging each figure to be more legible.

Response: Thank you very much for the heads up. We have enlarged each of the images in Figure 1 and Figure 2 as you suggested.

  1. Identify each specie comparison by figure (figure 3), makes for easier reading and quick comparison.

Response: Thank you very much for your reminder. We have clarified the species comparisons for each group as you suggested and made the changes in lines 202-207 of the article. Also, we have enlarged the words on the images as suggested.

  1. Identify each specie comparison by figure. Need larger images (figure4).

Response: Thank you very much for your reminder. We have clarified the species comparisons for each group as you suggested and made the changes in lines 223-227 of the article. Also, we have enlarged the words on the images as suggested.

Round 2

Reviewer 2 Report

1. If authors intented to compare chicken with other three meats, I highly recommend to chnage the title of this article. The title can mislead readers.

2. I suggest authors to check grammars and check singular and plural of nouns.

Author Response

The authors express sincere thanks to the your valuable comments and constructive suggestions.

1. If authors intented to compare chicken with other three meats, I highly recommend to chnage the title of this article. The title can mislead readers.

Response: Thank you for your advice. Considering that the article focuses on the metabolic differences between chicken meat and three other animal species (duck, pork, beef). We decided to change the title to ‘Metabolomics-based analysis of the major taste contributors of meat by comparing differences in muscle tissue between chickens and common livestock species’.

2. I suggest authors to check grammars and check singular and plural of nouns.

Response: Thank you for your advice. We have checked the grammar of the article, singular and plural of nouns and corrected the errors (line 123, line 162-164, line213-214, line 250, line292, line301, line 340, line 346-348).
